# Pre-Pandemic Dietary Assessment of Elderly Persons Residing in Nursing Homes—Silesia (Poland)

**DOI:** 10.3390/healthcare10050765

**Published:** 2022-04-20

**Authors:** Mateusz Grajek, Karolina Krupa-Kotara, Joanna Kobza, Antoniya Yanakieva

**Affiliations:** 1Department of Public Health, Faculty of Health Sciences, Medical University of Silesia in Katowice, ul. Piekarska 18, 41-902 Bytom, Poland; mgrajek@sum.edu.pl (M.G.); jkobza@sum.edu.pl (J.K.); 2Department of Epidemiology and Biostatistics, Faculty of Health Sciences, Medical University of Silesia in Katowice, ul. Piekarska 18, 41-902 Bytom, Poland; 3Department of Health Technology Assessment, Faculty of Public Health, Medical University Sofia, 1431 Sofia, Bulgaria; erasmus@foz.mu-sofia.bg

**Keywords:** COVID-19, SARS-CoV-2, elderly people, nursing homes, nutrition

## Abstract

Background. Residents of nursing homes (NHs) are one of the most vulnerable social groups to SARS-CoV-2 infection. It seems obvious that all preventive methods, including nutrition, should be a priority for these homes. Objective. The aim of this study was to evaluate the menus of elderly people residing in nursing homes and to compare them with the recommendations and especially with the scientific literature that proves the protective effect of nutrition on the course of COVID-19 disease. Material and methods. The material investigated in the research were decade menus selected at several nursing homes between 2017 and 2020. The total number of 4640 daily menus from 58 NHs located in Silesia (Poland) were analyzed in the research. Data analysis included mathematical tools of Kruskal–Wallis and U Mann–Whitney tests for multiple comparisons in scarcely observed samples (*p* = 0.05). Results. It has been noted that the energy value provided with food scored 1780.22 kcal, which denoted 102.72% of the daily standard for females and 98.23% for males. The investigated menus differed in terms of energy and nutrition value. The mean content of proteins totaled 47.95 g/day, which covered 93.83% of the daily requirements for this nutrient. When it came to fat content, a level of 109.12 g/day was observed; this covered 160.47% of the daily requirement for females and 143.58% for males. Absorbable carbohydrates constituted 116.60% of the daily standard, i.e., 151.59 g/day. It was stated that values for vitamin D reached 7.01 (±0.63) µg per day, which can be interpreted as 41.00% of the recommended intake for females and 42.00% for males. It was also noted that the values for vitamins A and E were respectively two and fifteen times lower than the recommendations. Conclusions. The evaluated menus must not be an aid in the prevention and treatment of COVID-19. The content of energy from food, fats, and carbohydrates substantially exceeded recommended standards, whereas the content of proteins, vitamins A, E, D, zinc and calcium did not meet requirements regarding nutrition standards for the analyzed group.

## 1. Background

For many years, the coronavirus infection has been considered harmless and associated with the occurrence of benign symptoms in the respiratory system, mainly in the upper airways [1]. Currently, with the emergence of highly infectious strains, which pose a real threat to the elderly, immunocompromised patients, and patients with chronic diseases, this view has changed [1]. The new type of coronavirus, which initially occurred only in China, quickly spread to other countries and eventually occurred on all continents. The dynamic development of the disease caused the World Health Organization to recognize COVID-19 as a pandemic on 11 March 2020 [2]. By the end of December 2020, COVID-19 has been diagnosed in more than 82 million people worldwide; nearly 2 million people died, and the mortality rate is estimated at 2.5% worldwide [3]. The SARS-CoV-2 virus, which belongs to the coronaviridae family, is an enveloping virus that is built of single-stranded RNA [4,5]. It is a completely new strain that has not been identified in humans until now [1]. Infection with this type of virus can be mild, with low intensity symptoms, or it can cause much more severe symptoms. In some cases, SARS-CoV-2 can lead to severe respiratory failure and, consequently, to multi-organ failure with possible mortality [5,6]. Unfortunately, COVID-19 is particularly dangerous for geriatric patients. The older age of patients is mentioned as an independent risk factor that increases mortality and contributes to the severe course of the disease [5,7]. Other risk factors include the presence of chronic disease especially ischaemic heart disease, diabetes, chronic obstructive pulmonary disease, chronic kidney disease, or hypertension [6]. According to data from the Chinese Centre for Disease Control and Prevention [8], the mortality from this disease for the 70–79-year-old age group is approximately 8%, and among people who are 80 years of age and older, it is as high as 14.8%. These values are much higher than the overall mortality rate, which in China is approximately 2.3% [8]. Similar data are obtained for the Italian population [5]. The overall mortality rate in Italy for COVID-19 is 7.2%; patients older than 70 years of age account for 37.6% of deaths. According to data from the Polish Ministry of Health, from the beginning of the pandemic in March 2020 to the end of December 2020, approximately 1.3 million people became ill with COVID-19 in Poland; more than 28 thousand people died, which means that the mortality among Poles is 2.2% [3]. The highest mortality rate is observed in the age group of people older than 60 years of age, who are also burdened with other diseases. The largest outbreak of the disease among the Polish population was in nursing homes (NHs), medical institutions, and hospital wards. In these places, the rapid spread of SARS-CoV-2 and the highest mortality rate are observed [3]. With the progression of the pandemic, the increasing nutritional awareness of society resulted in questions as to whether there are special dietary recommendations that will help avoid the infection or mitigate the course of the disease [9]. Unfortunately, thus far, none of the institutions and societies have issued explicit recommendations regarding this. The Polish Society of Dietetics has published a communication stating that there is no scientific basis and evidence for the use of specific food products, preparations, or vitamin and mineral supplements to prevent the SARS-CoV-2 infection [10]. At the same time, it has been suggested that a properly composed diet is sufficient to ensure the proper functioning of the immune system [11]. However, scientists pay special attention to the relation between the concentration of certain nutrients and the number of infections, the severity of the disease, and mortality owing to COVID-19 [12,13,14,15,16,17]; therefore, the continuous monitoring of nutritional status is recommended in patients who are confirmed to be infected with SARS-CoV-2. The European Society of Clinical Nutrition and Metabolism recommends that patients with a positive COVID-19 test result should be tested for malnutrition at the time of confirmation of the diagnosis owing to the risk of the rapid deterioration of health, which may lead to nutritional deficiencies and loss of muscle mass [18].

Due to the lack of unequivocal recommendations for daily rations in nursing homes, the quality of menus in terms of the content of individual nutrients recommended for the elderly is extremely important. Eating habits in nursing homes should meet the current standards for people aged 65+ (national standards) in terms of quality and quantity, i.e., basic needs of the body in terms of energy supply and micro- and macro-nutrients for a given age group. It should be remembered, however, that the nutrition standards used are standards for healthy people, and in Poland there are still no recommendations for NHs residents, which would take into account their health problems and the need for specific components (such as calcium, zinc and antioxidant vitamins). Applicable in the prevention of SARS-CoV-2 infections.

## 2. Aim

The aim of this study was to evaluate the menus of elderly people residing in nursing homes and to compare them with the recommendations (compared with standards based on the level of estimated average requirement (EAR) or adequate intake (AI). The adopted energy standard was determined based on the level of estimated energy requirements (EER) and especially with the scientific literature that proves the protective effect of nutrition on the course of COVID-19 disease.

## 3. Material and Methods

### 3.1. Study Area

The material investigated in the research were decade menus selected at several nursing homes between 2017 and 2020. Total number of 4640 daily menus (464 decade menus) from 58 NHs located in Silesia (Poland) were analyzed in the research (Figure 1).

### 3.2. Inclusion Criteria

The keeping of dietary records by the unit, i.e., annual dietary reporting in the form of decadal menus–is not so obvious in Poland, because menus are not a medical document according to the current legislation–some units asked to provide such data did not have/collect such data. Therefore, 58 units out of almost 80 similar units operating in the Silesia Province (Poland) entered the final accounts. In total, the following were analyzed: 4640 daily menus (464 decade menus).

### 3.3. Calculation Methods

To calculate the energy value and content of elementary micro- and macro-nutrients, the Dieta 5.0 computer program was used. The following values were estimated: energy [kcal], proteins [g], fats [g] and carbohydrates [g]–these data were collected to assess residents’ risk of malnutrition. The results obtained were compared with nutrition standards assessed for the Polish population [19] which referred to every healthy person (aged 66–75 years) with a standard body mass of 65 kg and physical activity level (PAL = 1.4). While elaborating the decisive results obtained, the loss value for vitamins of A–25.00%, B1–20.00%, B2–15.00%, C–55.00% was applied as a result of gastronomic processing. When it came to the remaining group of nutrients, the loss value totaled 10%. Recommendations for the Polish population were used for dietary fiber and cholesterol; they constituted: 20.00–40.00 g and <300.00 mg per day, respectively. The results obtained were compared with standards based on the level of the estimated average requirement (EAR) or adequate intake (AI). The assumed standard for energy was determined by using the group’s level of estimated energy requirement (EER) [19]. ORAC tables developed by the USDA were used to assess the antioxidant potential of food rations [20]. Although the values of antioxidants in the diet are not covered by specific standards, it is scientifically reported [21] that it is recommended to consume as many antioxidant rich foods as possible daily, which provide 5000.00 ORAC units expressed in μmol TE. In the qualitative evaluation of the menus, the Starzyńska, Bielińska, and Szewczyński methods [22] and the brief DQI (Diet Quality Index) questionnaire [23] were used.

### 3.4. Nutrition Standards

Due to the fact that in Poland there are no unified standards for nutrition of the elderly receiving institutional care, all values in the study were referred against the publication ‘Nutrition standards for the Polish population and their application’ [19], which was developed by the National Center for Nutrition Education functioning at the National Institute of Public Health-National Institute of Hygiene. All standards are in accordance with WHO recommendations. To help the reader interpret the results, a summary of the standards used is provided below. In selecting appropriate standards, the criteria described in section: ‘Calculation methods’ (Table 1) was used [19].

### 3.5. Statistical Analyses

With respect to statistical data treatment, descriptive statistics were executed for establishing the minimum and maximum value, numerical values for the arithmetic mean, median, mode, and standard deviation (SD). Data analysis included mathematical tools of the Kruskal–Wallis and U Mann–Whitney tests for multiple comparisons in scarcely observed samples with the strength of association coefficient epsilon-squared (E2). T-student test was used to compare whether the mean of each nutrient for each year was similar to the recommended dietary intake. Statistical significance level totalled *p* = 0.05. The analysis was conducted in the Statistica 13.0 computer program.

## 4. Results

It has been noted that the energy value provided with food scored 1780.22 kcal, which comprises 102.72% of the daily standard for females and 98.23% for males. The investigated menus differed in terms of energy and nutrition value. The mean content of proteins totaled 47.95 g/day, which covers 93.83% of the daily requirements for this nutrient. When it comes to fat content, the level of 109.12 g/day was observed; this comprises 160.47% of the daily requirement of females and 143.58% of males. Absorbable carbohydrates constituted 116.60% of the daily standard, i.e., 151.59 g/day (Table 2). It was shown that in each period represented, the value obtained from the decadal menus differed from the reference value indicated by the norm (was greater or smaller than it). This relationship was confirmed by the statistical tools used (F = 11.4681; *p* = 0.0001).

Statistical analysis has shown an adverse tendency concerning the overall fat content. This value has increased in individual years analyzed (*p* < 0.05). The above-mentioned tendency has not been stated for the remaining nutrients (Figure 2).

While analyzing the content of water-and fat-soluble vitamins in the assessed decade menus, it was stated that values for vitamin D reached 7.01 (±0.63) µg per day, which can be interpreted as 41.00% of the recommended intake for females and 42.00% for males. It was also noted that the values for vitamins A and E were respectively two and fifteen times lower than the recommendations. The content of micro-and macronutrients included in menus used at NHs exceeded the intake recommended for the following categories: sodium (2261.67 mg), phosphorus (1809.01 mg), magnesium (386.01 mg), iron (13.08 mg), and copper (0.51 mg). In terms of potassium and iodine content, the results obtained were 50% lower than the recommended intake. Recommended implementation for calcium was 85% of the standard, which means that 823.10 mg was included in decade menus. As the decade menus assessed did not verify the amount of salt added, the content of sodium and the implementation of its standards may not be precisely known (Table 3). In addition to that, supplementation with vitamins and minerals supplements did not take place at the analyzed NHs. The data concerning supplementation of residents’ diets were not collected. As in the case of macro-nutrients and in the case of vitamins and minerals, the values obtained from the analysis of the menus differed from the reference values (F = 10.3592; *p* = 0.0001).

The average ORAC value in all the menus analyzed was 3047.83 μmol TE (Table 4). The obtained value constituted 60.96% of the recommended intake indicated in the scientific literature.

Based on the methods used to assess the menus, it was shown that 86.84% of the acquired menus did not meet the criteria imposed in the applied evaluation methods (they were not suitable for improvement or contained numerous errors related to the variety of food products used). The analyzed decadence menus received the highest number of points. In Szewczyński’s method they scored34.49%, and the lowest score in Starzyńska’s method was 1.73% (Figure 3).

## 5. Discussion

The conducted study does have some limitations for example, the dietary documentation in the form of decade menus was reviewed, and this may not indicate the actual nutrition in these facilities, and it is uncertain whether the elderly actually ate the served portions, because it is not possible to check the so-called plate debris. Furthermore, it is unknown whether the meal was actually served. In addition, we are also not able to objectively determine the chemopreventive action of food in terms of SARS-CoV-2 prophylaxis. However, we have confidence in the people coordinating the nutrition in these institutions, and thus we assessed the menus using the indicated methods, which revealed a wide range of errors in these institutions. Additionally, the admission of the index PAL = 1.4 by Polish standards may not be objective and overstated for the elderly, often sick and with disabilities, requiring institutional care.

So far, there have been no similar studies carried out which assess decade-long diets in institutional nutrition, i.e., in the collective nutrition of people with special nutritional needs, which are undoubtedly the elderly. Moreover, the existing literature takes into account only illustrative works and is focused on the content of certain substances in the diets of the elderly. The correctness of the menus, according to which the seniors’ diet is later ordered, rarely works well on such a large scale. This also results in the indicated limitation of this publication, mainly related to this type of publication. Thus, it seems to be justified and does not affect the quality of the collected scientific evidence. Moreover, using inference by analogy, it can be assumed that proper nutrition will play a key protective function against various infectious diseases, including COVID-19.

The study brings out the fact that Poland still lacks management algorithms and guidelines in the nutrition of the elderly. The only ones available are based on the nutritional standards that are intended for healthy people. The aim of the study was to show the inefficiency of the Polish health care system and the lack of legal provisions in this area, which should be introduced immediately and implemented with the highest quality.

Referring to international data, the work provides an extensive literature review, also taking into account the aspect of the COVID-19 pandemic, which seems to be particularly important for the elderly population. Although this aspect was only an addition to this study, future studies will be extended to the aspects of mass nutrition in the prevention of SARS-CoV-2, because the authors have such data.

The results obtained in this study suggest that the planned menus at nursing homes do not fulfil their nutritional function in terms of quality and quantity. During this study, many discrepancies were noted between the values obtained based on the menus and the values recommended in the nutrition standards for the Polish population [19], and the recommendations from the scientific literature [21]. The study population is at risk of malnutrition, which is a direct cause of many adverse conditions, including increased risk of infection (e.g., SARS-CoV-2). Immunity is defined as the entirety of defense mechanisms that provide protection against pathogens that cause disease in the body. Nutritional status plays a key role in the immunomodulatory effect in the course of viral infections, especially in the elderly. The process of phagocytosis, activation of cytokines, and reduction of oxidative stress by fighting free radical molecules contribute to the pro-health spectrum of the immune system. This process can be stimulated by biologically active compounds present in nutrients. Enriched food or a balanced diet taking into account vitamins and minerals: C, A, E, zinc and especially increased daily intake of cholecalciferol-vitamin D in the form of a supplement, contribute to the fight and prevention of the multiplication of SARS-CoV-2 virus cell particles, and minimize the mortality rate and complications resulting from the course of the COVID-19 disease. Specifically, the high supply of energy, fats, and carbohydrates and the insufficient supply of proteins, vitamin D, and calcium should be taken into account as well as the low diversity of menus, which may translate into their low antioxidant status. Many scientific studies [12,13,14,15,16,17,24,25,26,27,28,29,30,31,32,33,34,35,36,37,38,39,40] have shown that a properly composed diet may greatly affect the body’s immunity and, consequently, protective action against the SARS-CoV-2 infection [24]. Special attention is paid to the relationship between the vitamin D concentration and the number of infections, the severity of the disease, and mortality due to COVID-19 [12]. Vitamin D has pleiotropic effects; receptors for this vitamin have been found in almost all cells including those of the immune system. 25OH-D stimulates cytokines, accelerates immune response, and has an anti-inflammatory effect, which may be particularly important for the SARS-CoV-2 infection [25]. In Switzerland, a cohort study has been conducted to evaluate the relationship between the SARS-CoV-2 test result and the plasma concentration of vitamin D. The presence of SARS-CoV-2 was tested by PCR using a nose and throat swab. The analysis of the results showed that positive COVID-19 cases had a significantly lower median 25OH-D than negative COVID-19 cases [26]. Some scientists have also suggested that vitamin D levels affect the severity of COVID-19 [27]. Observations performed in selected European countries confirm that there is a negative relationship between the average serum concentration of 25OH-D and the number of COVID-19 cases per million inhabitants; the same relationship occurs between vitamin D levels and the course of the disease and the number of COVID-19 deaths [28]. However, the effect of 25OH-D on the frequency and severity of infection with a new type of coronavirus remains unclear and requires additional studies. However, according to the current recommendations, vitamin D supplementation is recommended [25]. In our study, it was determined that the vitamin D content in decadent menus intended for the residents was six times lower than the current recommendations. Moreover, some studies [29,30,31] have shown that vitamin D combined with zinc and vitamin C is an integral part of the immune system and shows synergistic functions at different stages of defense such as in maintaining the integrity of biological barriers and the functionality of cells that form the immune system. Therefore, a deficiency of these key components may lead to the damage of the mucosal epithelial cells and possibly make them more susceptible to the penetration of pathogens such as SARS-CoV-2. In our study, the value of zinc in menus was 13 times lower than the recommended value. Other studies, including Zhou et al.’s [32] indicate some affinity between the body’s calcium level and the course of the SARS-CoV-2 infection. In a clinical study, mild and moderate cases showed unsatisfactory levels of calcium in the early stages of viral infection, while severe and critical cases showed significantly lower levels of calcium than mild and moderate cases in the early stages. It was also determined that low levels of calcium were associated with multi-organ damage in the course of COVID-19. The analysis of the study material from the decadence menus showed that the calcium value was 823.10 mg, which was 85% of the recommended intake for this component. Another important aspect of the nutrition of the elderly is the supply of foods rich in antioxidant compounds in the diet. In our study, the antioxidant potential of foods was assessed based on ORAC tables [20,21]. It was shown that the average value of this index in all studied periods was 3047.83 μmol TE, which according to Prior [21] represents 60.69% of the recommended value (5000 μmol TE). Of note, in our study, the value of antioxidant vitamins A and E was, respectively, two and fifteen times lower than the value indicated in the standards. Many current scientific reports indicate that the antioxidants contained in the diet and dietary supplements can greatly affect the course of COVID-19. Biancatelli et al. [33] in their studies have shown the simultaneous administration of quercetin and ascorbic acid is an experimental strategy for the prevention and treatment of infection with certain respiratory viruses (such as SARS-CoV-2). Blocking virus penetration is the key strategy, and quercetin hinders the fusion of viral membranes both for influenza [34] and SARS-CoV and MERS-CoV in vitro studies [35]. In addition, other studies [36,37,38,39,40,41] have shown that many polyphenols can be inexpensive and safe prophylaxis that reduce virus infectivity and the risk of the cytokine storm. At the molecular level, polyphenols act as viral protease inhibitors that participate in viral replication owing to their general affinity for proteins through hydrogen binding and their low risk of toxic effects. The same may be true for the binding of these compounds to the S-protein (virus spike). General recommendations for the elderly and NHs residents during the COVID-19 pandemic should focus on pro-healthy dietary patterns [42]. They can be generally described as rich in plant products, including fresh fruit and vegetables, soybean, nuts, good sources of antioxidants [43] and omega-3 fatty acids [44], as well as a low content of saturated and trans fats, animal protein, and added/refined sugars [45]. Most of these dietary goals can be achieved with a traditional Mediterranean diet [46], which is rich in protective and anti-inflammatory polyphenols [47]. The necessity to change the diet of elderly people depends on physiological, sociological, economic, and customary factors. Malnutrition may occur even when a person has access to an adequate amount of food with a wide variety of assortments. In the older age, the saliva secretion decreases, the loss of taste buds leads to a reduction in taste sensations, dry mouth, difficulties in the formation of bites, and thus may reduce consumption to a greater extent than required by physiological changes. Many studies show the association of both quantitative and qualitative malnutrition on the development of many diseases, as well as the risk of infection [48,49]. Malnutrition consists of an inadequate supply of major nutrients such as proteins, fats, and carbohydrates, but also minerals and vitamins, which have been extensively described above. In view of this, it is necessary to identify solutions that would give NHs the basis to be able to properly arrange menus aimed at the elderly. In addition to country-specific standards, various dietary models and dietary programming approaches are useful, e.g., Linear Programming proposed by Corné van Dooren [50] and the Emerging Programming Model proposed by Luca Benvenuti and Alberto De Santis [51]. The authors would like to emphasize that in the future, when conducting similar studies, nutrition monitoring should be carried out, because it will make them error-free The lack of data on plate debris, i.e., the amount that the patient actually consumed, was a limitation of this study. Based on this study, we can see how important it is to analyze and evaluate the quality of nutrition in aid institutions.

In addition, knowledge about the nutrition of the elderly should be widely disseminated, especially in institutions providing this nutrition, and should not be available and discussed only in scientific circles.

Retrospective cross-sectional studies, therefore, may become a tool of health education, which is a method of primary prevention which isimportant in the elderly population in the face of the COVID-19 pandemic. Future research should focus on the influence of nutrition on the functioning of the immune system and on identifying target subgroups of populations with the most sensitive immune systems such as the elderly and people with chronic diseases.

## 6. Conclusions

Based on the research available, it has been stated that the assessed menus were badly arranged regarding energy and nutrition:The content of energy from food, fats, and carbohydrates substantially exceeded recommended standards, whereas the content of proteins did not meet requirements regarding the nutrition standards for the analyzed group.When it came to vitamins and both micro- and macro-nutrients, a low supply of A, E, D vitamins, zinc and calcium was observed.The analyzed menus were characterized by a small variety, which translates into their low evaluation in the methods used and insufficient antioxidant potential.

Considering the above, the menus under review cannot be helpful in the prevention of SARS-CoV-2 infections and the treatment of COVID-19.

## Figures and Tables

**Figure 1 healthcare-10-00765-f001:**
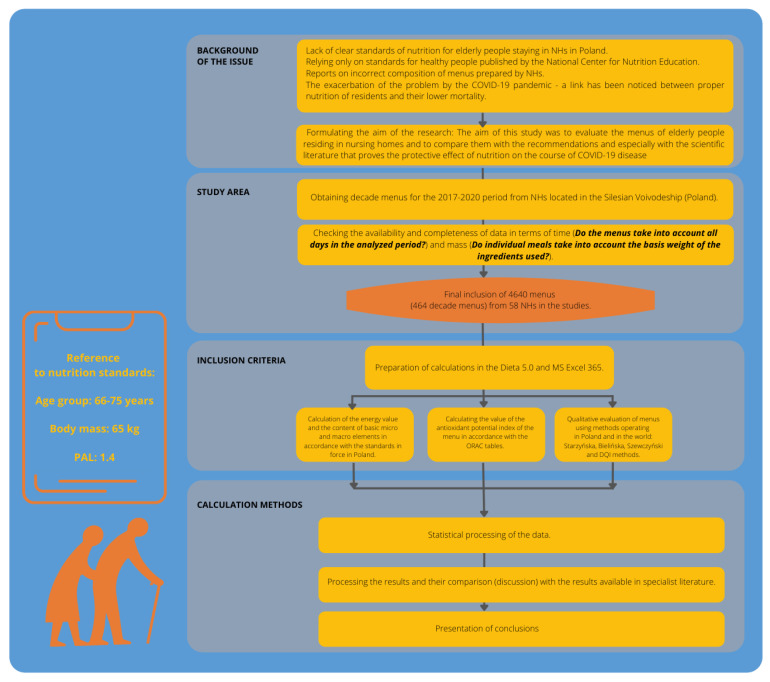
Publication search algorithm.

**Figure 2 healthcare-10-00765-f002:**
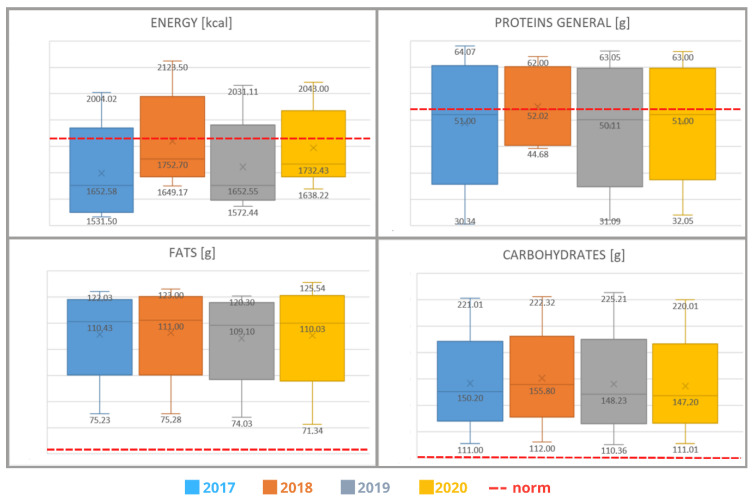
The content of selected nutrients in decade menus.

**Figure 3 healthcare-10-00765-f003:**
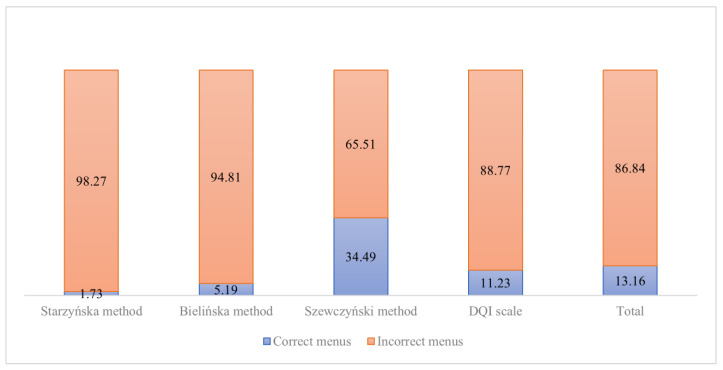
Qualitative evaluation of menus.

**Table 1 healthcare-10-00765-t001:** Nutrition standards used in the study.

Component	Limit of Standard [19].	
	Female	Male
**Energy (kcal)**	1750.0	1900.0
**Protein general (g)**	52.0	52.0
**Fats (g)**	59.0	63.0
**Carbohydrates (g)**	45.0–65.0	45.0–65.0
**C (mg)**	60.0	75.0
**B1 (mg)**	0.9	1.1
**B2 (mg)**	0.9	0.9
**B3 (mg)**	11.0	12.0
**B6 (mg)**	1.3	1.4
**B9 (µg)**	320.0	320.0
**A (µg)**	50.0	63.0
**D (µg)**	15.0	15.0
**E (mg)**	7.0	9.0
**K (µg)**	115.0	115.0
**Na (mg)**	1300.0	1300.0
**K (mg)**	3800.0	3800.0
**Ca (mg)**	1000.0	1000.0
**P (mg)**	580.0	580.0
**Mg (mg)**	265.0	350.0
**Fe (mg)**	6.0	6.0
**Zn (mg)**	7.0	9.0
**Cu (mg)**	0.3	0.3
**I (µg)**	95.0	95.0

**Table 2 healthcare-10-00765-t002:** Energy and nutrition value of assessed decade menus and standards implementation.

Assessed Year		Statistics					Standards Implementation (%)	
		Minimum Value	Maximum Value	Average (SD)	Median	Mode	Females	Males
**Energy (kcal)**	**2017**	1531.50	2004.01	**1690.32** 84.31	1652.58	1543.04	**101.31**	**97.41**
	**2018**	1720.23	2123.50	**1810.11** 103.53	1752.52	1733.08	**108.50**	**100.03**
	**2019**	1620.31	2031.11	**1780.22** 80.11	1652.52	1543.43	**103.26**	**99.23**
	**2020**	1731.01	2043.00	**1764.98** 94.21	1732.43	1721.10	**102.34**	**98.31**
	**Total**	1619.45	2094.00	**1780.22** 95.19	1752.55	1733.00	**102.72**	**98.23**
**Proteins in general (g)**	**2017**	30.34	64.07	**48.87** 5.57	51.00	52.76	**94.34**	
	**2018**	29.04	62.00	**47.56** 6.21	52.02	51.78	**93.53**	
	**2019**	31.09	64.01	**46.99** 6.45	50.11	50.34	**92.91**	
	**2020**	32.05	63.05	**47.01** 5.66	51.00	50.88	**93.17**	
	**Total**	31.03	63.00	**47.95** 6.37	50.10	51.62	**93.83**	
**Fats (g)**	**2017**	75.23	122.03	**101.11** 6.44	110.43	111.55	**150.31**	**133.18**
	**2018**	75.28	123.00	**105.12** 2.21	111.00	108.00	**155.65**	**138.76**
	**2019**	74.03	120.30	**110.34** 8.67	109.10	110.44	**158.37**	**149.85**
	**2020**	71.34	125.54	**114.15** 7.51	110.03	111.06	**161.47**	**151.13**
	**Total**	76.11	123.83	**109.12** 7.55	112.70	113.00	**160.47**	**143.58**
**Carbohydrates (g)**	**2017**	111.00	221.01	**149.23** ±14.91	150.20	141.00	**114.23**	
	**2018**	112.00	222.32	**150.01** 10.99	155.80	140.46	**115.68**	
	**2019**	110.36	225.21	**151.53** 17.32	148.23	141.14	**116.57**	
	**2020**	111.01	225.07	**148.98** 15.23	147.20	138.00	**113.98**	
	**Total**	113.00	220.01	**151.59** 17.39	149.00	142.00	**116.60**	

**Table 3 healthcare-10-00765-t003:** The content of selected vitamins and minerals in assessed menus (daily intake).

Vitamins/ Minerals	Assessed Year					Standards Implementation (%)	
	2017	2018	2019	2020	Average (SD)	Females	Males
**C (mg)**	112.20	114.23	115.40	116.01	**116.02** 12.1	**193.00**	**154.00**
**B1 (mg)**	3.02	4.30	5.45	1.01	**2.21** 0.12	**174.00**	**143.00**
**B2 (mg)**	1.00	5.00	2.20	3.01	**2.02** 0.33	**245.00**	**207.00**
**B3 (mg)**	17.45	18.00	20.00	22.00	**19.33** 2.10	**196.00**	**180.00**
**B6 (mg)**	1.00	3.20	2.00	6.00	**2.45** 0.12	**199.00**	**184.00**
**B9 (µg)**	339.10	342.34	341.00	348.00	**345.34** 12.05	**102.00**	**108.00**
**A (µg)**	150.23	158.00	151.89	151.00	**151.45** 34.36	**312.00**	**257.00**
**D (µg)**	8.02	10.04	9.00	9.60	**7.01** 0.63	**41.00**	**42.00**
**E (mg)**	16.01	18.02	20.08	22.59	**18.34** 3.01	**284.00**	**246.00**
**K (µg)**	92.34	100.30	96.42	91.02	**98.45** 8.05	**85.00**	**88.00**
**Na (mg)**	2258.02	2261.89	2261.09	2266.02	**2261.67** 23.07	**176.00**	**187.00**
**K (mg)**	1986.45	1986.30	1986.05	1986.00	**1986.40** 6.08	**48.00**	**48.00**
**Ca (mg)**	819.22	820.00	823.00	830.00	**823.10** 3.01	**850.00**	**850.00**
**P (mg)**	1805.00	1802.00	1815.33	1810.00	**1809.01** 68.23	**311.00**	**311.00**
**Mg (mg)**	382.45	388.60	390.00	386.70	**386.01** 12.03	**140.00**	**101.00**
**Fe (mg)**	15.00	18.33	12.30	11.00	**13.08** 3.41	**218.00**	**218.00**
**Zn (mg)**	14.01	12.00	18.05	15.80	**13.01** 1.12	**198.00**	**139.00**
**Cu (mg)**	0.40	0.48	0.58	0.52	**0.51** 0.01	**158.00**	**181.00**
**I (µg)**	41.02	50.00	43.22	56.67	**49.09** 0.24	**51.00**	**51.00**

**Table 4 healthcare-10-00765-t004:** ORAC values for analyzed menus.

**ORAC** **(μmol TE)**	**Assessed Year**	**Minimum Value**	**Maximum Value**	**Average** **(SD)**	**Median**	**Mode**
	**2017**	1575.03	5015.45	**3170.83** 635.71	3079.03	3079.55
	**2018**	1685.20	5125.34	**3280.83** 614.11	3189.34	3189.65
	**2019**	1884.33	5324.21	**3479.83** 634.20	3388.23	3388.11
	**2020**	1687.01	5127.11	**3282.83** 623.01	3191.11	3191.12
	**Total**	1452.08	4892.23	**3047.83** 625.27	2956.71	2956.34

## Data Availability

Not applicable.

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
