# Peer review of "Pre-Pandemic Dietary Assessment of Elderly Persons Residing in Nursing Homes—Silesia (Poland)"

_healthcare, 2022, doi:10.3390/healthcare10050765_

Round 1
Reviewer 1 Report
The present study entitled: "Assessment of the Diet of Elderly People in Nursing Homes in 2 the Face of the COVID-19 Pandemic" is a retrospective cross-sectional study in which dietary evaluations of the foods offered in different Nursing Homes were carried out and making comparisons of food contributions through the years prior to the COVID-19 pandemic, as well as comparisons with dietary recommendations for the target population. The work is original and of great importance and relevance for the nutrition of the elderly, however, the work is written with a bias towards the importance of nutrition in the COVID-19 disease and needs revisions that allow the body to focus of the article towards the proposed objective.
The title of the article is not very explanatory, I suggest that it be changed considering the main objective and without emphasizing COVID-19, because nothing that has to do with the disease is being evaluated, it could be placed as referring to a period as “pre-pandemic evaluation for COVID-19”, for example.
The objective of the study is not clear between the abstract and what is placed in the text, it should be clarified if the objective was only to evaluate the menus or make comparisons with the recommended daily intakes.
The study design must be placed at the beginning of the materials and methods section. We suggest that the authors adhere to the STROBE guidelines for reporting observational studies, so that they meet the minimum reporting requirements.
The authors report that the values ​​of the nutritional standards for the Polish population were obtained, we suggest the authors to place the reference values ​​used in the tables, this to make it easier for readers to understand the implementation standards. In the same way, we suggest that comparative analyzes be applied to the samples of each year with the standards for each nutrient, this can be done with a t-test for one sample. This would allow knowing if the average of each nutrient for each year was like the recommended dietary intake.
Another suggestion is to present the data as mean and confidence interval, rather than standard deviation, because it is easier to interpret if the averages for each year have an estimate similar to the recommended daily intake.
The use of the symbol ± is not recommended when presenting standard deviation data, it is only useful in the presentation of standard errors, it should be changed to the symbol “SD:”
The authors should indicate what each of the units presented in figure 1 mean. I imagine they are the averages, but it would be useful if the authors presented these data in a dot-and-bar error plot instead of a bar graph, since it is more useful for the interpretation of results, they could present the values ​​as mean and confidence intervals for each nutrient in each year. In the same way, it would be suggestive that the authors put the p values ​​of the comparisons between years.
The discussion of the work is focused on the dietary aspects that can be related to the prevention or better management of COVID-19, it is an important review work, unfortunately the work presented has little or nothing to do with food during COVID-19, the authors should restructure their discussion to lead the reader to interpret their results with respect to their objective, which is written in this way: “the main aim of this study was to assess the dietary habits at nursing homes in terms of their energy value and content of selected micro- and macronutrients”. At no time is food mentioned during COVID.19, I understand the importance of this work for the improvement of the dietary quality of the canteens that could improve the nutritional status which is related to a poor immune status, but the relationship of The entire discussion, and also the introduction with COVID-19, is forced, mainly because there is no real data on what has happened during the pandemic in the years 2020 and 2021. The authors should also add comparisons of their results with similar studies in your country or in the world, so that readers have more information about it.
Author Response
Dear Reviewer,
Thank you very much for the insightful analysis of our manuscript and the submitted comments. Each of the comments was analyzed and used appropriately.
Response 1: Title has been corrected (1-5)
Response 2: The aim has been corrected (14-16) and (100-108)
Response 3: The study design is presented as a diagram in Figure 1. (114-116)
Response 4: In order not to change the configuration of the current tables and to facilitate the perception of the reception, the table (156) has been added
Response 5: It was corrected as suggested (177-180, 201-203).
Response 6: Has been corrected as suggested (204).
Response 7: It was corrected as suggested (185).
The study addresses an important, but often underestimated, topic of nutrition of elderly people living in nursing homes, especially in the face of the aging Polish society, which is especially important in the time of the COVID-19 pandemic.
The limitation of the study is the fact, that in the conducted study the dietary documentation in the form of decade menus was reviewed, which may not indicate actual nutrition in these facilities, and whether the elderly actually ate the served portions, because it is not possible to check the so-called plate debris, and whether the meal has actually been served. In addition, we are also not able to objectively determine the chemopreventive action of food in terms of SARS-CoV-2 prophylaxis. However, we have confidence in the people coordinating nutrition in these institutions, and thus we assessed the menus using the indicated methods, which revealed a wide range of errors in these institutions. Additionally, the admission of the index PAL = 1.4 by Polish standards may not be objective and overstated for the elderly, often sick and with disabilities, requiring institutional care.
So far, there was no similar studies carried out, to assess decade-long diets in institutional nutrition, i.e. in collective nutrition of people with special nutritional needs, which are undoubtedly the elderly. Moreover, the existing literature takes into account only illustrative works and is focused on the content of certain substances in the diet of the elderly. The correctness of the menus, according to which the seniors' diet is later ordered, rarely works well on such a large scale. This also results in the indicated limitation of this publication, mainly related to this type of publication. Thus, it seems to be justified and does not affect the quality of the collected scientific evidence. Moreover, using the inference by analogy, it can be assumed that proper nutrition will play a key protective function against various infectious diseases, including COVID-19.
The study brings out the fact that Poland still lacks management algorithms and guidelines in the nutrition of the elderly. The only ones available are based on the nutritional standards that are intended for healthy people. The aim of the study was to show the inefficiency of the Polish health care system and the lack of legal provisions in this area, which should be introduced immediately and implemented with the highest quality.
Referring to international data, the work covers an extensive literature review, also taking into account the aspect of the COVID-19 pandemic, which seems to be particularly important for the elderly population. Although this aspect was only an addition to this study, the next studies will be extended to the aspects of mass nutrition in the prevention of SARS CoV-2, because the authors have such data.
The authors would like to emphasize in the future that when carrying out similar studies, nutrition monitoring should be carried out, thanks to which they will be error-free, in the form of the lack of data on plate debris, i.e. the amount that the patient actually consumed. Based on this study, we can see how important it is to analyze and evaluate the quality of nutrition in aid institutions.
In addition, knowledge about the nutrition of the elderly should be widely disseminated, especially in institutions providing this nutrition, and should not be available and discussed only in scientific circles.
Retrospective cross-sectional studies, therefore, may become a tool of health education, which is a method of primary prevention, so important in the elderly population in the face of the COVID-19 pandemic.
Thank you very much for all comments and observations. We hope that, after taking into account the changes, our manuscript will meet the substantive expectations, and the corrections made will be sufficient to publish the article in Healthcare. The review is extremely valuable to us and provides us with an instruction, on how to properly edit articles of a similar nature in the future.
Best regards, Karolina Krupa-Kotara

Reviewer 2 Report
The research makes an important contribution to the literature. After taking into account a few corrections, the work does not raise any objections.
- The abstract is well organized and contains the most relevant information.Introduction - correct.
- I suggest that the methodology section be improved and that the authors break it down into more specific subchapters. There is a lack of description of some important parts, namely the inclusion / exclusion criteria from the research.
- Data analysis and results are correct.
- The discussion is well written. I suggest that the authors further develop the implications of this study for I propose to add the limitations of the study (strengths and weaknesses).
- The list of bibliographic references is current and correct.
I don't feel qualified to judge about the English language and style.
Congratulations on your research
Author Response
Dear Reviewer,
Thank you very much for the insightful analysis of our manuscript and the submitted comments. Each of the comments was analyzed and used appropriately.
Response 1: Keeping dietary documentation by the unit, i.e., keeping 5-year nutrition reporting in the form of decade menus, is not obvious in Poland, because menus are not a medical document in the light of applicable legal regulations and some units asked to provide such data simply did not had it. Therefore, 58 units from nearly 80 similar units operating in the ÅšlÄ…skie Voivodeship (Poland) were included in the final settlements. The chapter "Material and methods" has been reorganized and the missing elements have been supplemented (109-169).
Response 2: Thank you for your comment, the inclusion criteria, strengths and limitations of the study were taken into account (218-252, 346-358).
The study addresses an important, but often underestimated, topic of nutrition of elderly people living in nursing homes, especially in the face of the aging Polish society, which is especially important in the time of the COVID-19 pandemic.
The limitation of the study is the fact, that in the conducted study the dietary documentation in the form of decade menus was reviewed, which may not indicate actual nutrition in these facilities, and whether the elderly actually ate the served portions, because it is not possible to check the so-called plate debris, and whether the meal has actually been served. In addition, we are also not able to objectively determine the chemopreventive action of food in terms of SARS-CoV-2 prophylaxis. However, we have confidence in the people coordinating nutrition in these institutions, and thus we assessed the menus using the indicated methods, which revealed a wide range of errors in these institutions. Additionally, the admission of the index PAL = 1.4 by Polish standards may not be objective and overstated for the elderly, often sick and with disabilities, requiring institutional care.
So far, there was no similar studies carried out, to assess decade-long diets in institutional nutrition, i.e. in collective nutrition of people with special nutritional needs, which are undoubtedly the elderly. Moreover, the existing literature takes into account only illustrative works and is focused on the content of certain substances in the diet of the elderly. The correctness of the menus, according to which the seniors' diet is later ordered, rarely works well on such a large scale. This also results in the indicated limitation of this publication, mainly related to this type of publication. Thus, it seems to be justified and does not affect the quality of the collected scientific evidence. Moreover, using the inference by analogy, it can be assumed that proper nutrition will play a key protective function against various infectious diseases, including COVID-19.
The study brings out the fact that Poland still lacks management algorithms and guidelines in the nutrition of the elderly. The only ones available are based on the nutritional standards that are intended for healthy people. The aim of the study was to show the inefficiency of the Polish health care system and the lack of legal provisions in this area, which should be introduced immediately and implemented with the highest quality.
Referring to international data, the work covers an extensive literature review, also taking into account the aspect of the COVID-19 pandemic, which seems to be particularly important for the elderly population. Although this aspect was only an addition to this study, the next studies will be extended to the aspects of mass nutrition in the prevention of SARS CoV-2, because the authors have such data.
The authors would like to emphasize in the future that when carrying out similar studies, nutrition monitoring should be carried out, thanks to which they will be error-free, in the form of the lack of data on plate debris, i.e. the amount that the patient actually consumed. Based on this study, we can see how important it is to analyze and evaluate the quality of nutrition in aid institutions.
In addition, knowledge about the nutrition of the elderly should be widely disseminated, especially in institutions providing this nutrition, and should not be available and discussed only in scientific circles.
Retrospective cross-sectional studies, therefore, may become a tool of health education, which is a method of primary prevention, so important in the elderly population in the face of the COVID-19 pandemic.
Thank you very much for all comments and observations. We hope that, after taking into account the changes, our manuscript will meet the substantive expectations, and the corrections made will be sufficient to publish the article in Healthcare. The review is extremely valuable to us and provides us with an instruction, on how to properly edit articles of a similar nature in the future.
Best regards, Karolina Krupa-Kotara
